# Relationships between Parent-Reported Parenting, Child-Perceived Parenting, and Children’s Mental Health in Taiwanese Children

**DOI:** 10.3390/ijerph16061049

**Published:** 2019-03-23

**Authors:** Ching-Yu Huang, Yi-Ping Hsieh, April Chiung-Tao Shen, Hsi-Sheng Wei, Jui-Ying Feng, Hsiao-Lin Hwa, Joyce Yen Feng

**Affiliations:** 1Department of Psychology, Bournemouth University, Fern Barrow BH12 5BB, UK; 2Department of Social Work, University of North Dakota, Grand Forks, ND 58202, USA; yiping66@gmail.com; 3Department of Social Work, National Taiwan University, Taipei 106, Taiwan; acshen@ntu.edu.tw (A.C.-T.S.); yencertain@gmail.com (J.Y.F.); 4Department of Social Work, National Taipei University, New Taipei City 23741, Taiwan; hswei@mail.ntpu.edu.tw; 5Department of Nursing, National Cheng Kung University, Tainan City 701, Taiwan; juiying@mail.ncku.edu.tw; 6Department and Graduate Institute of Forensic Medicine, National Taiwan University, Taipei 100, Taiwan; hwahl013@ntu.edu.tw

**Keywords:** parenting, culture, gender differences, child mental health

## Abstract

The current study examines the relationship between parents’ and children’s reports of parenting and their effects on children’s mental health symptoms. Six hundred and sixty-six parent-child dyads in Taiwan participated in this study. The parents and the children filled out the parenting questionnaires, and the children also reported their general mental health. The results demonstrated that parental-reported and child-perceived parenting were positively correlated, but parents tended to report lower scores on authoritarian parenting and higher scores on Chinese parenting than did their children. There were also significant gender differences: The mothers reported higher authoritative parenting than did the fathers; and the boys perceived higher authoritarian and Chinese-culture specific parenting than did the girls. Moreover, the Chinese parenting had a negative effect on children’s mental health outcomes. Finally, our results showed that children’s perception of parenting had a stronger effect on children’s mental health symptoms than did parental reports on parenting, urging future research to include the children’s report when investigating the effects of parenting on children’s mental health outcomes.

## 1. Introduction

Parenting affects various aspects of children’s psychological health, such as self-esteem [1], emotional regulation [2], socio-emotional adjustment, and well-being [3]. The majority of parenting research have adopted two major approaches: a dimensional approach which focuses on individual dimensions of parental behaviors (such as control/demandingness and warmth/responsiveness) [4], and a categorical approach that categorizes parenting according to a combination of parenting dimensions into parenting styles, such as authoritative, authoritarian, permissive, or neglectful parenting styles [5]. Parental responsiveness/warmth can be characterized by supportive, sensitive, accepting, and nurturing parental behaviors, while parental demandingness/control has been conceptualized as a set of active parental strategies involving the communication of clear and consistent expectations for appropriate behavior and efforts to monitor the child’s behavior related to these expectations [6]. Using the two parenting dimensions, four parenting styles can be identified: authoritative parenting characterized by high levels of parental warmth as well as control; authoritarian parenting with low levels of parental warmth and high degree of parental control; permissive parenting with high levels of parental warmth yet low levels of parental control, and lastly, neglectful parenting with minimal levels of parental warmth and control. Studies among European American populations have yielded consistent results demonstrating the association between authoritative parenting style and positive child outcomes, including better self-esteem and better mental health [7,8]. Close parent–child relationships characterized by warmth and acceptance are often protective against the development of children’s depression and conduct problems [9,10]. In contrast, harsh parenting practice or the authoritarian parenting style have been associated with increased problem behaviors and depressive symptoms [10,11].

However, parenting styles and beliefs are subjected to social and cultural influences [12,13,14], and the effects of different parenting styles on children’s developmental outcomes also vary across cultural groups [15]. In Chinese and Chinese immigrant populations, similar to their European American counterparts, authoritative parenting has been associated with positive child adjustment [16,17,18], whereas associations between authoritarian parenting and child outcome has been mixed. Some studies have found that harsh or authoritarian parenting has predictable negative consequences for both European American and Chinese children, including emotion dysregulation [19], aggression [20], lower educational attainment [21], and depressive symptoms [22,23] and children’s behavioral deviance [18,24]. Conflicting results showed that Chinese children were generally satisfied with their parents’ authoritarian parenting, and they perceived their parent–child relationships more positively if their parents were more authoritarian [25]. Therefore, authoritarian parenting may have less negative effects among Chinese children (as opposed to their Euro-American counterparts) because they view parents’ attempts to regulate them as an act of love [26]. High parental endorsement of “training”, an indigenous Chinese parenting concept dictating parents teaching children early through guidance and monitoring of their behaviors, while providing care, concern, support, and parental involvement [26], has been shown to reduce the correlation between authoritarian parenting and both internalizing and externalizing problems among Chinese-American immigrant children [20], suggesting the discipline–behavior problem link being moderated by cultural context. These findings have led researchers to advocate for a culturally anchored approach to understanding and classifying parenting styles.

Although authoritarian and authoritative parenting styles are both found in Chinese societies [16,24] and immigrant Chinese parents [17,27], some culturally important and specific Chinese parenting concepts cannot be fully captured using parenting typologies constructed in European American cultures [26,28]. For instance, Kim et al. [22] used latent profile analyses on eight parenting dimensions to identify four parenting profiles among Chinese American parents: supportive (high on both parental warmth and positive control), tiger (high on both parental warmth and hostility), easygoing (low on both parental warmth and hostility), and harsh (low on parental warmth and high on negative control) parenting. Their results demonstrated the supportive parenting profile being the most common, and it was associated with the best developmental outcomes. The second most common was the easygoing parenting, followed by tiger parenting and harsh parenting. Tiger parenting was not the most typical parenting profile in Chinese American families, nor did it lead to optimal developmental outcomes among Chinese American adolescents [22]. Moreover, researchers have identified five Chinese-culture specific parenting styles beyond the widely-accepted authoritative and authoritarian parenting styles in Chinese and immigrant Chinese parents: Parental protection, directiveness, shaming, encouragement of modest behavior, and maternal involvement [14].

In Chinese culture, young children are viewed as incapable of understanding and making decisions that are in their own best interests [29]. Therefore, ‘*parental protection*’ reflects Chinese parents’ expectations of themselves having the responsibility to provide a safe environment for their children. Such responsibilities are seen as the primary responsibility of parents of young children [14]. Based on the same assumption that young children are incapable of understanding, ‘directiveness’ refers to parents taking major responsibility for regulating their children’s behavior and academic performance [30]. ‘Shaming’ is a Chinese socialization practice that helps children learn to be sensitive to the perceptions, feelings, evaluations, and judgments of others in order to teach them to avoid future behaviors that would bring shame or embarrassment to the family [31]. ‘Encouragement of modest behavior’ refers to encouraging children to behave in a humble and socially conforming way when interacting with others to maintain social and interpersonal harmony, because maintaining such harmony is of prime importance in traditional Chinese society [32]. ‘Maternal involvement’ describes Chinese mothers’ extensive involvement and devotion to their children, particularly during the early years, by providing a very nurturing environment for the children by being physically available and by promptly attending to all their children’s needs [14].

These five Chinese parenting constructs were shown to not overlap with and were independent of the authoritative and authoritarian constructs identified in North American research [14]. Chinese mothers reported using more Chinese parenting than their U.S. and U.K. counterparts [14,33]; whereas the Chinese immigrant mothers in the U.S. and the U.K. reported similar level in the Chinese-specific parenting to their non-immigrant counterpart in Taiwan [34]. These findings demonstrated that these Chinese parenting practices are still prevalent in Chinese populations, including Chinese immigrants. Although some of the Chinese cultural-specific parenting values, such as shaming and parental protection, were found to be associated with children’s internalizing symptoms [22,35], research examining the effect of Chinese culture-specific parenting on children’s mental health remains scarce. Therefore, one of the objectives of the current study is to examine the relationships between different parenting styles, particularly Chinese parenting, and their effects on children’s mental health outcome.

Other than the ethnic and cultural issues in this line of research, another methodological constraint in parenting research lies in the source of the parenting reports. Some researchers studied the association between parent and child reports of parenting and found that these were only low to moderately correlated [36,37]. Previous research did find associations between parenting practices and child externalizing behaviors [11] as well as internalizing symptoms [10]; the majority of research in this area relied on parental reports of parenting practices, especially with younger children [38,39]. The child’s perspective on how their parents affect their emotional and behavioral functioning had been less extensively examined [36], perhaps due to questions concerning the reliability and validity of such reports. Fortunately, increasing research has shown that adolescents can provide reliable and valid reports of parenting and their own emotional and behavioral functioning [8,9,40,41]. Children may be influenced by their perceptions of parental behaviors rather than by actual parental behaviors or those reported by the parents [42]. Indeed, empirical studies demonstrated that conclusions based on parents’ reports are quite different from those derived from children’s reports on parallel measures [43], and agreements between parental and child reports on parenting practices varied according the aspects of parenting examined [37,44]. Moreover, parents from different socio-cultural contexts may have different behavioral norms as well as parenting expectations, thus it is not surprising to see cultural background as one of the moderating factors affecting the agreements between parental and child reports on parenting [37]. Indeed, past research reported a high level of discrepancy between the parenting practices Chinese American adolescents experience and those reported by their parents [45]. Therefore, including both parents’ and children’s reports of parenting in examining effects of parenting on child mental health will provide valuable information. Thus, another objective of the current study is to examine the associations between parental reports and children’s perception of parenting and compare their effects on children’s psychological symptoms.

Previous research has established the important concurrent and long-term influence of parenting on children’s development and psychological health. However, the links between parents’ and children’s reports on parenting, and each of their effect on children’s mental health outcomes, had rarely been examined within the same study despite concerns regarding congruences between parental and child reports having been evident [37,43,44]. Furthermore, research regarding the effect of cultural contexts and culturally-rooted parenting practice alongside the more established authoritative and authoritarian parenting on child mental health was even rarer within such discussions, especially within the Chinese cultural context. Thus, the current study will be the first of its kind in examining both parents’ and children’s reports on parenting, especially including the Chinese cultural-specific parenting, on children’s mental health within the Chinese population. The first objective of the current study is to examine whether the parent and the children report differently on the following three parenting styles: authoritative, authoritarian, and Chinese parenting, which are situated at a similar level psychometrically. The second objective is to compare the effects of parental reported parenting and child-reported parenting on children’s psychological symptoms. We aim to answer the following questions: (1) Do parents’ reports of parenting differ from the children’s perception of parenting? If so, how do they relate to one another in each parenting dimension? (2) Does children’s perception of parenting account for more variance than does parental reported parenting in children’s psychological symptoms? Based on previous research findings, we hypothesized that there will be significant differences between parental and children’s reports on each parenting dimension, particularly with parents reporting higher levels of authoritative parenting than their children. We also hypothesized that children’s perception of parenting will explain more variance in children’s psychological symptoms than will parental reports of parenting.

## 2. Materials and Methods

### 2.1. Participants and Procedure

The participants were 666 fourth-grade (mean age = 10.4 years, SD = 0.50, range = 9.2 to 10.8) Taiwanese students, with 310 boys (46.5%) and 356 girls (53.5%), and their parents (mean age = 39.7 years, SD = 5.38, with 180 (27%) fathers and 486 (73%) mothers). Most (580, 87%) of the parents were married or cohabitating; some parents were divorced/separated (73, 11%) or in single-parenthood (13, 2%). The monthly income of these families (measured by self-report) ranged from less than 1333 USD/month (245, 36.8%), between 1334 and 2665 USD/month (259, 38.9%), to over 2666 USD/month (154, 23.1%). Eight (1.2%) of the families refused to answer the question regarding monthly family income. Eighty-eight (13.2%) of the parents obtained 12 years or less formal education, 442 (66.3%) had vocational or high school diploma, and 136 (20.4%) held a bachelor or graduate degree.

This project was the pilot study of a larger research project approved by the National Taiwan University Hospital Institutional Review Board (approval code 201305042RINB). In addition to the formal project IRB approval, this pilot study was reviewed and approved by the advisory board committee members of the National Taiwan University Children and Family Research Center. After gaining ethical approval, we first stratified the sample by geographical locations (i.e., 6 counties or cities in northern, southern, and central Taiwan) before we randomly selected primary schools in each county or city for recruitment. Trained research assistants contacted the selected schools’ principals for recruitment. The consenting schools distributed both a cover letter and an informed consent form to the parents of the 4th-grade (10 years old) students. The cover letter and consent form clearly explained the objectives of the research project, emphasizing the voluntary nature of the study, and the contact details of the research team were also given. Once the parents consented to participate, they received the parenting questionnaires to complete before the data collection. Only 17.3% of the parents of the participating schools agreed to participate, and the parents self-identified as the primary caregiving parent of the child. Due to constraints of resources and time, we were only able to include the primary caregiving parents for participation in the study. Their children then brought the signed informed consent forms and questionnaires in sealed envelopes back to the classroom where the research assistants retrieved them on the day of the data collection for the children. Nearly all (99.9%) of the children of the consented parents agreed to participate. Before letting the students sign the informed consent forms, our research assistants explained the study’s purpose and procedures to students, emphasizing the voluntary and anonymous nature of the research. Then, the research assistants distributed self-report questionnaires to consenting students in group sessions at the time agreed with the schools, either during or outside of regular class hours. After completing the questionnaires, the participating students received a set of stationery, and participating parents received 100 TWD (equivalent of 3.3 USD) gift voucher as a token of our thanks.

### 2.2. Measures

Parenting. Parents reported their parenting styles using the Parenting Styles and Dimensions Questionnaire (PSDQ) [14,46]. We used the version from Wu et al., as this version had already been validated in the Chinese population [14]. The questionnaire covers 3 parenting subscales: the authoritative parenting (15 items, such as, “I give praise when my child is good” and “I give comfort and understanding when my child is upset“), the authoritarian parenting (11 items, example items included “I yell or shout when my child misbehaves” and “I spank when my child is disobedient”), the Chinese-specific parenting (18 items, for instance “I overly worry about my child getting hurt” and “I tell my child that I get embarrassed when he/she does not meet my expectations”). Wu et al. [14] had excluded the permissive subscale from the original PSDQ questionnaire because it could not be measured reliably in Chinese samples. All items were rated on a 5-point scale ranging from 1 (never) to 5 (always). Mean scores of each subscale were used for subsequent statistical analyses. The internal consistency of the parenting questionnaire (Cronbach’s α) demonstrated high internal consistency for each subscale (α = 0.919 for authoritative parenting subscale, α = 0.870 for authoritarian parenting subscale, and α = 0.757 for Chinese parenting subscale).

Perceived parenting. We adapted the questionnaire items from Wu et al. [14]’s version of Parenting Styles and Dimensions Questionnaire [14,46] for it to be appropriate for the child participants. The questionnaire’s subscales cover 3 perceived parenting styles: the authoritative parenting (4 items, such as, “My parents give praise when I am good” and “My parents give comfort and understanding when I am upset“), the authoritarian parenting (3 items, example items included “My parents yell or shout at me when I misbehave” and “My parents spank me when I am disobedient”), and the Chinese-specific parenting (3 items, for example, “My parents overly worry about me getting hurt” and “My parents tell me that they get embarrassed when I do not meet their expectations”). The questionnaire was modified in order to measure the same parenting constructs as the original PSDQ whilst shortened significantly in order to fit the child population. Items were selected based on their factor loading, so only the highest loading items were chosen for each dimension. All items were rated on a 5-point scale anchored by 1 (never) and 5 (always). Mean scores of each dimension were used for subsequent statistical analyses. The perceived parenting questionnaire also demonstrated high internal consistency (Cronbach’s α = 0.839 for perceived authoritative parenting subscale, 0.746 for perceived authoritarian parenting subscale, 0.574 for perceived Chinese parenting subscale). In order to ensure the validity of the perceived parenting scale, a Confirmatory Factor Analysis with varimax rotation was conducted, and the results confirmed that the three-factor structure (authoritative, authoritarian, and Chinese parenting) was maintained for the perceived parenting scale (see Table 1). The estimated model chi-square was χ^2^ (18, *N* = 666) = 52.962, *p* < 0.001. The Comparative Fit Index (CFI) was 0.946; the Tucker–Lewis Fit Index (TLI) was 0.972, and the RMSEA was 0.054 (with confidence interval between 0.046 and 0.061), demonstrating acceptable fit for the model.

The Brief Symptom Rating Scale (BSRS-5). The Brief Symptom Rating Scale (BSRS-5) [47] is composed of 5 self-report items by which participating children can evaluate their psychological symptoms in the past week. The BSRS-5 is commonly used in Taiwan for screening psychological disorders and has been shown to significantly predict healthy participants’ quality of life, demonstrating empirical support for external validity [48]. It measures anxiety (e.g., “I felt tense or high-strung”), hostility (e.g., “I felt easily annoyed or irritated”), depression (e.g., “I felt depressed or in a low mood”), interpersonal sensitivity (e.g., “I felt inferior to others”), and additional symptoms (e.g., “I had trouble falling asleep”). The score for each item ranges from 0 to 4 (0 = not at all, 1 = a little bit, 2 = moderately, 3 = quite a bit, and 4 = extremely), and the sum score of these five items were used in the subsequent analyses. The internal consistency of the BSRS-5 (Cronbach’s α) was 0.876, demonstrating high internal consistency. A sum score of the BSRS-5 above 14, or a score of more than 1 on the additional suicide survey item, may indicate a severe mood disorder. Scores between 10 and 14 may indicate moderate mood disorders, and those between 6 and 9 could indicate mild mood disorders [48]. The participants with BSRS-5 scores lower than 5 were considered to be normal [47].

### 2.3. Statistical Analyses

We used SPSS version 23 for data analyses. First, descriptive statistics were used to assess the distribution of parent-reported parenting, child-perceived parenting, and mental health. Second, we conducted correlational analyses to illustrate interrelationships between each of the variables. Thereafter, a multivariate analyses of covariance (MANCOVA) using parent and child gender as independent variables, monthly family income as a covariate, and parent-reported parenting, child-perceived parenting, and child mental health as dependent variables to assess the effect of parental and child gender on parenting and child mental health. Gender differences in parenting were examined because past research [49,50] suggested parenting can be affected by parental as well as child gender. The correlational analyses and the MANCOVA were used to identify possible confounding factors so we can control for their effects in the final hierarchical regression analyses. Paired-sample t-tests were conducted to test for differences between parent-reported parenting and child-perceived parenting. Finally, we conducted hierarchical regression to examine the effects of parent-reported parenting and child-perceived parenting on child mental health after controlling for the effect of family income and child and parental gender.

## 3. Results

### 3.1. Descriptive Statistics and Correlations

Table 2 presents descriptive statistics of the means and standard deviations of the variables, and Table 3 presents the correlation coefficients among all the independent and dependent variables. Because monthly family income was positively correlated with parent-reported authoritative parenting (Spearman’s ρ (649) = 0.133, *p* < 0.01) as well as child-perceived authoritative parenting (ρ (648) = 0.167, *p* < 0.01) but negatively correlated with parent-reported authoritarian parenting (ρ (647) = −0.091, *p* < 0.05), we decided to control for the effects of monthly family income in subsequent analyses. Parents’ marital status did not have significant effects on parents’ reported parenting, children’s perceived parenting, or children’s psychological symptoms (examined using MANOVA); therefore, parents’ marital status was not included in the subsequent analyses.

### 3.2. Gender Effect on Parent-Reported Parenting, Child-Perceived Parenting, and Child Mental Health

The effects of parental and child gender were examined as part of the preliminary analyses. A two-way MANCOVA was conducted to examine the effects of child and parental gender on child-mental health and parent-reported and child-perceived parenting dimensions while controlling for the effects of monthly family income. The MANCOVA revealed significant main effects of parental gender (F_(1, 615)_ = 9.223, Pillai–Bartlett trace= 0.095, *p* < 0.001, η^2^ = 0.095) and child gender (F_(1, 615)_ = 3.809, Pillai–Bartlett trace= 0.042, *p* < 0.001, η^2^ = 0.042) on child mental health. Follow-up univariate analyses of variance (ANCOVAs) with Bonferroni corrections were then conducted to examine the effects of monthly family income and parental and child gender on each of the dependent variables. The ANCOVA revealed significant effects for monthly family income (covariate) on parent-reported authoritative parenting (F_(1, 621)_ = 13.398, *p* < 0.001), parent-reported authoritarian parenting (F_(1, 621)_ = 4.530, *p* < 0.05), and child-perceived authoritative parenting (F_(1, 621)_ = 15.077, *p* < 0.01).

Results also revealed significant univariate effects for child gender on child-perceived authoritative parenting (F_(1, 621)_ = 10.660, *p* < 0.001) and child-perceived authoritarian parenting (F_(1, 621)_ = 19.371, *p* < 0.001) as well as a significant univariate effect for parental gender on parent-reported authoritative parenting (F_(1, 621)_ = 34.842, *p* < 0.001). There was no significant interaction effect between child and parental gender. Subsequent pairwise comparisons revealed that boys perceived their parents using more authoritarian (MD = 0.262, *p* < 0.01) and Chinese-culture specific (MD = 0.372, *p* < 0.001) parenting than did girls; and mothers reported higher authoritative parenting (MD = 0.317, *p* < 0.01) than did fathers. Because child gender and parental gender had significant effects on child-perceived and parent-reported parenting respectively, we controlled for their effects in the subsequent hierarchical regression analyses.

### 3.3. Differences between Parent-Reported and Child-Perceived Parenting

Paired-sample t-tests were used to examine differences between parent-reported parenting and child-perceived parenting across three parenting dimensions: authoritative, authoritarian, and Chinese parenting. The results revealed that parents reported significantly lower scores on authoritarian parenting than did their children (*t*(657) = −5.09, *p* < 0.001) but significantly higher scores on Chinese parenting than did their children (*t*(656) = 10.06, *p* < 0.001; see Table 4 for summary of the t-tests).

### 3.4. Effects of Parent-Reported Parenting and Child-Perceived Parenting on Child Mental Health

The hierarchical multiple regression analyses (see Table 5) examined whether parent-reported authoritative, authoritarian, and Chinese parenting were significant contributors to children’s mental health symptoms, after covariates and the effect of child-perceived parenting were controlled for. Control variables (monthly family income, child gender, and parental gender) were entered into the regression model in the first step (Model 1). Child-perceived parenting variables (i.e., child-perceived authoritative, authoritarian, Chinese parenting) were entered in the second step (Model 2). Parent-reported parenting variables (i.e., parent-reported authoritative, authoritarian, Chinese parenting) were entered as the third step (Model 3).

The results showed that family monthly income, child gender, and parental gender jointly explained 0.4% of the variance (R^2^ = 0.004, F_(3, 621)_ = 0.834, *p* = 0.475), and the model was not significant. Adding child-perceived parenting significantly increased the proportion of variance explained (∆F_(3, 618)_ = 20.624, *p* < 0.001; ∆R^2^ = 0.091) and adding parent-reported parenting did not significantly increased the proportion of variance explained (∆F_(3, 615)_ = 1.909, *p* = 0.127; ∆R^2^ = 0.008).

The regression coefficients indicated that child-perceived authoritative parenting significantly decreased child psychological symptoms scores, whereas child-perceived authoritarian and Chinese parenting significantly increased child psychological symptoms scores even after the effects of child gender, parental gender, and family income were controlled for.

In summary, our findings demonstrated that although parents’ and children’s reports on parenting were generally consistent (positive but low correlations), parents tended to under-report their use of authoritarian parenting and over-report their use of Chinese-specific parenting compared to their children. As for gender difference, boys perceived their parents using more authoritarian and Chinese-culture specific parenting than did girls; and mothers reported higher authoritative parenting than did fathers. Our findings from the hierarchical regressions showed that children’s perception of parenting was a stronger predictor of children’s mental health symptoms than parental reports on parenting, urging future research to include children’s reports when investigating the effects of parenting on children’s mental health outcomes.

## 4. Discussion

Consistent with previous research [36,51,52], our findings demonstrated low agreement between parents’ and children’s report on parenting. As expected, parents tend to present their childrearing behaviors more favorably, whereas children report less healthy patterns of family functioning than their parents [36,51]. Similar to findings from Euro-American populations [36,52], the Taiwanese parents under-report their use of authoritarian parenting compared to their children. The Taiwanese parents’ higher reported scores in Chinese parenting than their children may reflect the fact that Taiwanese parents endorse the Chinese culture-specific parenting values much more than their younger generation. It might also be that the children are not enculturated enough to pick up on the culture-specific practice from their parents.

With regards to the association between parenting and child mental health, our results are consistent with previous research [19,20,21] that the mental health of Taiwanese children, just like their Euro-American counterparts, is also negatively affected by authoritarian parenting and positively affected by authoritative parenting. The negative effect of Chinese parenting on Taiwanese children’s mental health is novel and alarming. Previous research showed that shaming, one of the dimensions measured in the Chinese parenting style, overlaps somewhat with Euro-American notions of psychological control [53,54] and could threaten children’s self-esteem and increase internalizing problems in Euro-American society [35]. This helps to explain the effect of Chinese parenting style on children’s psychological symptoms, suggesting that some aspects of Chinese parenting have undesirable consequences on children’s mental health. Although some earlier research on Chinese population has suggested that authoritarian parenting may not have as harmful an effect on child outcomes [25,26], our findings corroborate more recent research [20,23], suggesting that authoritarian parenting have similar negative influence on child mental health outcomes in Chinese populations. Some aspects of Chinese parenting, such as parental protection, if done excessively (over-protection), may have negative consequences on children’s mental health, such as depression [23] or internalizing symptoms [10,55]. The impacts of different aspects of Chinese parenting should be further examined in future research. Moreover, our findings from the regression analyses showed that after children’s perception of parenting was accounted for, parental report on parenting did not increase the variance explained in child mental health, suggesting children’s mental health is influenced more by their perceptions of parental behaviors, rather than by actual parental behaviors or those reported by the parents [42,56,57]. Previous research also demonstrated that the effects of parenting on children’s mental health can be mediated by children’s trait emotional intelligence [58] and children’s experience of exposures to violence [59], underscoring the importance of the roles children themselves and their experiences play in mediating the impacts from their parents’ parenting.

As for effects of gender, our results showed that mothers reported higher authoritative parenting than fathers did; and boys perceived their parents’ parenting to be more authoritarian and Chinese-culture specific than girls did, but parents did not differ in their reported parenting for boys and girls. The parental gender effect was consistent with recent findings that mothers tend to report more authoritative parenting than do fathers [50,60,61]. This is also consistent with the traditional Chinese parental roles: ‘strict father, warm mother’, portraying fathers as more authoritarian, controlling, and strict than mothers, who are portrayed as nurturing and supportive [62]. Although child gender did not affect the parents’ self-reported parenting, boys did report higher perceived authoritarian parenting than did girls, which is consistent with previous findings that boys tend to be disciplined more harshly than girls [21,49,63]. Although fathers’ and mothers’ reports on their own and each other’s parenting are generally consistent, they also have a unique influence on child outcomes [50,61,64,65]. Therefore, it is advisable in future studies to include parenting reported by both fathers and mothers to examine their joint and unique contribution to child outcomes. Additionally, findings on the gender differences may also be explained by parent–child communication, where adolescent girls tend to be more interdependent of their parents, thus experiencing better quality relationships with their parents than do boys [66,67]. Future studies should incorporate the potential effects of parent–child communication in the examination of parenting and children’s wellbeing.

The current study provides unique insights into the relationships between parental and child reports of parenting as well as their effects on children’s mental health in the Taiwanese population. However, some limitations need to be acknowledged. First, the current study’s design is cross-sectional, making it difficult to determine the direction of influence. It is also possible that children who are depressed tend to interpret their parents’ parenting less favorably and more harshly. Future studies with longitudinal designs are needed to further verify such relationships and directions of influence. Second, our parenting measures included only the primary caregiving parent’s self-report and child report. Adopting a multi-informant approach, such as including both parents’ report and observer’s report to study parenting, would provide a better assessment of parenting [49,59]. Moreover, the internal consistency for perceived Chinese parenting subscale was relatively low, and thus our findings will need to be validated by future research. Due to constraints on time and resources, we could not explore the diverse components of Chinese parenting in more detail, which surely warrants more attention in future research. Third, child psychological symptoms were only measured by children’s self-report. Future studies should include report from different sources, such as parents and teachers, to provide more information. Finally, the current study only included participants from Taiwan, limiting the generalizability of the findings. Even within Chinese ethnic groups, the sub-cultures across different Chinese societies (such as Mainland China, Hong Kong, Taiwan, Macau) could be considerably distinct [52]. Future research including non-Chinese samples as well as Chinese populations from various societies would further our understanding of the interaction between parenting and child mental health in different social-cultural contexts.

## 5. Conclusions

The current study provided insights into the relationships between parental and child reports of parenting as well as their effects on children’s mental health, adding the Chinese cultural perspectives to further our understanding of parenting and child mental health outcome. Our findings also remind scholars and clinicians as well as policy makers on the importance of taking children’s perspectives into account when examining their mental health outcome, as they may have more direct influence on their mental health outcomes. For instance, multiple sources could be considered when assessing children and adolescents’ adjustment, but the voices from the young people should be especially valued. In the Chinese socio-cultural context, parents tend to be especially involved in their children’s education. However, the parents’ overprotection may have undesirable effects on children’s mental health. Thus, education professionals should be especially aware of such a delicate balance between parental involvement and respecting children’s own voice. In addition, given the links between parental approval of physical punishment and child physical abuse [68,69], the Taiwanese children, especially boys, perhaps are at a higher risk of physical abuse, and this risk might be perpetuated by the promotion of parental authority in Chinese culture. The undesirable effect of authoritarian and Chinese parenting on children’s mental health warrants more future investigations with a Chinese population, underscoring the importance of looking at human development from a holistic and culturally-sensitive perspective.

## Figures and Tables

**Table 1 ijerph-16-01049-t001:** Factor loadings with varimax rotation of each item in the perceived parenting scale.

Perceived Parenting Scale Items	Factor 1 Authoritative	Factor 2 Authoritarian	Factor 3 Chinese
My parents give praise when I am good	**0.829**	−0.031	−0.046
My parents give comfort and understanding when I am upset	**0.841**	−0.169	−0.017
My parents encourage me to talk about my troubles with them	**0.797**	−0.093	0.071
My parents explain and discuss the consequences of my behavior with me	**0.774**	−0.028	0.109
My parents spank me when I am disobedient	−0.043	**0.821**	0.070
My parents slap me when I misbehave	−0.051	**0.777**	0.090
My parents yell or shout at me when I misbehave	−0.158	**0.769**	0.136
My parents overly worry about me getting hurt	0.341	−0.182	**0.598**
My parents tell me that they get embarrassed when I do not meet their expectations	−0.104	0.352	**0.625**
My parents make me feel guilty when I do not meet their expectations	−0.045	0.254	**0.793**

Note: Factor loading greater than 0.400 were highlighted in bold.

**Table 2 ijerph-16-01049-t002:** Mean, standard deviations, and range of variables.

Measures	Mean	SD	Range
Parent-reported Authoritative Parenting	3.85	0.62	1–5
Parent-reported Authoritarian Parenting	2.02	0.55	1–5
Parent-reported Chinese Parenting	2.94	0.45	1–5
Child-perceived Authoritative Parenting	3.89	0.94	1–5
Child-perceived Authoritarian Parenting	2.21	0.95	1–5
Child-perceived Chinese Parenting	2.57	0.89	1–5
Child Psychological Symptoms			
Anxiety	0.69	0.91	0–4
Hostility	0.86	1.01	0–4
Depression	0.57	0.90	0–4
Interpersonal Sensitivity	0.78	1.02	0–4
Additional Symptoms	0.50	0.88	0–4
BSRS-5 total score	3.38	3.77	0–20

**Table 3 ijerph-16-01049-t003:** Pearson Correlation coefficients among parent-reported parenting, child-perceived parenting, and child psychological symptoms.

Variables	1.	2.	3.	4.	5.	6.
1. Authoritative parenting (P)	—					
2. Authoritarian parenting (P)	−0.270 ***	—				
3 Chinese parenting (P)	0.121 **	0.275 ***	—			
4. Authoritative parenting (C)	0.195 ***	−0.125 **	0.024	—		
5. Authoritarian parenting (C)	−0.060	0.265 ***	0.105 **	−0.209 ***	—	
6. Chinese parenting (C)	−0.036	0.117 **	0.141 **	0.090 *	0.330 ***	—
7. Psychological symptoms (C)	−0.002	0.121 **	0.124 **	−0.234 ***	0.216 ***	0.129 **

Note: (P) Parental report, (C) Child report; * *p* < 0.05, ** *p* < 0.01, *** *p* < 0.001, 2-tailed.

**Table 4 ijerph-16-01049-t004:** Differences between parent-reported and child-perceived parenting across all parenting dimensions.

Parenting Styles	Parent-Reported		Child-Perceived		*t*	*p*	95% CI	
	M	SD	M	SD			LL	UL
Authoritative parenting	3.85	0.62	3.89	0.94	−0.922	0.357	−0.114	0.041
Authoritarian parenting	2.01	0.55	2.20	0.95	−5.091 ***	0.000	−0.265	0.118
Chinese parenting	2.93	0.45	2.57	0.89	10.055 ***	0.000	0.297	0.441

*** *p* < 0.001.

**Table 5 ijerph-16-01049-t005:** Results of hierarchical multiple regressions predicting child psychological symptoms.

Variables	Child psychological Symptoms					
	Model 1		Model 2		Model 3	
	*β*	Beta	*β*	Beta	*β*	Beta
Model 1 Control variables						
Child Gender	−0.337	−0.045	0.030	0.004	0.023	0.003
Parental Gender	0.344	0.041	0.327	0.039	0.226	0.027
Family Income	0.027	0.011	0.119	0.048	0.133	0.053
Model 2 Child-perceived parenting						
Authoritative			−0.894	−0.221 **	−0.913	−0.226 **
Authoritarian			0.540	0.138 **	0.485	0.124 **
Chinese			0.414	0.100 *	0.373	0.090 *
Model 3 Parent-reported parenting						
Authoritative					0.160	0.032
Authoritarian					0.177	0.032
Chinese					0.408	0.077
Model Summary	R^2^ = 0.004		ΔR^2^ = 0.091		ΔR^2^ = 0.008	
	*F*_(3, 621)_ = 0.834 (n.s.)		*F* Δ _(3, 618)_ = 20.624 **		*F* Δ _(3, 615)_ =1.909 (n.s.)	

* *p* < 0.05, ** *p* < 0.01.

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
