# Peer review of "Relationships between Parent-Reported Parenting, Child-Perceived Parenting, and Children’s Mental Health in Taiwanese Children"

_ijerph, 2019, doi:10.3390/ijerph16061049_

Round 1
Reviewer 1 Report
This is an improved version of the previous version of the reviewed paper and the authors have done their best to address all the raised issues.
Author Response
Thank you very much for your approval and contribution.
Reviewer 2 Report
In my opinion, this manuscript has been improved by authors. Although this manuscript has some limitations, I think it provides some interesting results about how different parenting styles are related to children’s mental health in Taiwanese children.
In the revised manuscript, authors have clarified some important aspects about the instrument used to measure parental styles. In this regard, the clarification that they have made about the use of the PSDQ version from Wu et al. is important; indicating that this version had already been validated in Chinese population.
If the authors have also used this version of Wu et al. to measure the perceived parenting in children (adapting the questionnaire items of this scale), it should be indicated in the manuscript (lines 212-213): “Perceived parenting. We adapted the questionnaire items from the Parenting Styles and Dimensions Questionnaire [14,46] for this measure to be appropriate for the child participants.”
Also, it would be convenient to review some sentences, in which the dimensions of authoritarian and authoritative parenting, and Chinese parenting are situated at a similar level (line 150): “first objective of the current study is to examine whether the parent and the children report differently on the following 3 dimensions of parenting: authoritative, authoritarian, and Chinese parenting”.
In my opinion, after reviewing the aspects indicated above, this manuscript could be published.
Author Response
Thank you very much for the thoughtful suggestions.
We have added in line 214-215 that the Perceived Parenting measure was adapted from Wu et al’s (2002) version.
We have also modified the sentence in line 151-152 to incorporate the excellent point they made about these parenting styles situating at a similar level.
This manuscript is a resubmission of an earlier submission. The following is a list of the peer review reports and author responses from that submission.
Round 1
Reviewer 1 Report
This was a very interesting paper to review and the under investigating factors were clearly explained in a feasible and robust study. The reviewed literature which was presented in the introduction part presented a firm base for the hypothesis which the authors were trying to examine. The adopted methodological approach sounds perfect and the presented findings were sufficient enough to justify the information presented in the discussion part.
Although English is my second language and I am not completely qualified to judge the English style of information presentation, I should say that except for some mismatch between the used tense and styles of information presentation (i.e. line 215 to 217: Because child gender and parental gender had significant effects on child-215 perceived and parent-reported parenting respectively, we will control for their effects in the 216 subsequent hierarchical regression analyses)no other problems with English usage noticed.
I also find it very useful to see some examples of the three types of parental styles which were under investigation in this study. Two or three examples form each style will give a general impression regarding the overall style of parenting in each subtype.
I am so keen on knowing more about the marital status of the parents who participated. Were all of them couples or there were also single parents who participated in this study. I also think that data on parental marital status and comparing it with the style that they are adopting might be very important.
I also think that the authors wisely finalized the paper through applicability of the findings for different groups who might benefit getting insight on the reported relationships between parents and children impression of parenting, as well as their effects on children’s mental health in a Chines population.
Author Response
Dear Reviewer,
First of all, we sincerely thank you for your very helpful and positive feedback for our paper.
We are very thankful for all the constructive suggestions you have made.
1. In our revision, we have corrected the tense in line 436-437 to make it consistent (all in past tense now).
2. Two example items from each parenting dimension were added to illustrate them in the method section (2.2 measures, lines Lines 321-326 and 334-339).
3. The descriptive statistics on parents' marital status were added in the description of sample, but there were no significant differences among parents with different marital status regarding their parenting practice or their children's perceived parenting (examined using MANOVA). A line reporting the null result of parents' marital status was added in Result section 3.1 (Line 395-397).
Again, we would like to thank you for your very positive and constructive feedback for our paper!
Reviewer 2 Report
I appreciate the opportunity to review the paper titled “Relationships Between Parent-reported Parenting, Child-perceived Parenting and Children’s Mental Health in Taiwanese Children. This study addresses an important topic, parenting and children's mental health, using a sample of Taiwanese students, with 310 boys (46.5%) and 356 girls (53.5%), and their parents, with 180 (27%) fathers and 486 (73%) mothers. The strength of this papers is that it examines differences between parent-reported parenting and child-perceived parenting. The weakest part of this paper is the theoretical fundament of the study and on how this study can contribute to existing theory. What is the importance of this manuscript?
Below the authors may find as follows a few specific comments relating to each of the sections:
1) Introduction. The introduction section could be improved still further. Authors have based the design of the manuscript in old references. I have only read two recents references:
1. Huang, C.Y.; Lamb, M.E. Acculturation and Parenting in First-Generation Chinese Immigrants in the United Kingdom. Journal of Cross-Cultural Psychology 2015, 46, 150-167. doi:10.1177/ 0022022114555763
2.Huang, C.Y.; Cheah, C.S. L.; Lamb, M.E.; Zhou, N. Associations Between Parenting Styles and Perceived Child Effortful Control Within Chinese Families in the United States, the United Kingdom, and Taiwan. Journal of Cross-Cultural Psychology 2017, 48(6), 795–812. doi: 10.1177/0022022117706108.
Therefore, this problem affects the overall manuscript. Authors should search recent issues of scientific literature to see if their topics have been covered recently. They can include in the revision results from meta-analyses about parenting studies. Then, they should include several current studies to rewrite the introduction section. Nowadays, there are many articles that consider the adolescents' point of view on parenting like:
1.García, O. F., Serra, E., Zacarés, J. J., & García, F. (2018). Parenting styles and short-and long-term socialization outcomes: A study among Spanish adolescents and older adults. Psychosocial Intervention, 27(3), 153-161.
2. Martínez, I., Garcia, F., Fuentes, M. C., Veiga, F., Garcia, O. F., Rodrigues, Y., ... & Serra, E. (2019). Researching Parental Socialization Styles across Three Cultural Contexts: Scale ESPA29 Bi-Dimensional Validity in Spain, Portugal, and Brazil. International journal of environmental research and public health, 16(2), 197.
3. Moed, A., Gershoff, E. T., & Bringewatt, E. H. (2017). Violence exposure as a mediator between parenting and adolescent mental health. Child Psychiatry & Human Development, 48(2), 235-247.
4. Moreno-Ruiz, D., Estévez, E., Jiménez, T., & Murgui, S. (2018). Parenting style and reactive and proactive adolescent violence: evidence from spain. International Journal of Environmental Research and Public Health, 15(12), 2634.
5. Tabak, I., & Zawadzka, D. (2017). The importance of positive parenting in predicting adolescent mental health. Journal of Family Studies, 23(1), 1-18.
2) Introduction. The authors did not explain important classic parenting studies, such as those conducted by Baumrind or Maccoby & Martin.
3) Introduction. In the introduction section the authors wrote (line 61, 62): “some culturally important and specific Chinese parenting concepts cannot be fully captured using parenting typologies constructed in European American cultures [20,22].” and (line 71, 72, 73, 74): “Moreover, researchers have identified five Chinese-culture specific parenting styles beyond the widely-accepted authoritative and authoritarian parenting styles in Chinese and immigrant Chinese parents: Encouragement of modest behavior, Parental protection, Shaming, Directiveness, and Maternal involvement [9].” However, authors did not analyze five Chinese-culture specific parenting styles separately. Why? I believe this point is very important because I find Chinese-culture specific parenting styles are interesting and emerging research.
4) Introduction. Can the authors formulate hypothesis? Thus, the objective would be much clearer.
5) Participants. It would be interesting to see more detail around the sample and the procedures. Here, more information are required. Authors should explain clearly the age range of the students.
6) Participants. Authors have not justified that there were more mothers than fathers. This fact can distort the results that would be obtained with a balanced sample by sex.
7) Measures. The authors wrote (line 149, 150, 151, 152): “The questionnaire was modified in order to measure the same parenting constructs as the original PSDQ whilst shortened significantly in order to fit the child population. Items were selected based on their factor loading, so only the highest loading items were chosen for each dimension. Perceived parenting.” Why was not a confirmatory factorial analysis carried out to know if the structure of the instrument is maintained?
8) Measures. Authors did not describe examples of the items.
9) Measures. The Chinese parental dimension also generates several doubts. Why not analyze the different dimensions separately?
10) Results. I have detected some problems with scores of The Brief Symptom Rating Scale (BSRS-5). It appears that many children scored really low on psychological symptoms. Therefore, it would be difficult to talk about differences in symptomatology. There was a great social desirability or the instrument has problems to discriminate between children.
Authors should try to find a new angle that has not been covered adequately in the previous study, and incorporating new material that has accumulated in recents years. These suggestions may change the results and discussion of the manuscript. I hope that the authors find these comments useful and that they improve the manuscript.
Author Response
Responses to Reviewer 2:
We thank Reviewer 2 for their meticulous examination and constructive feedback.
Please see our responses detailed in the following:
1. We thank the reviewer for the suggestion for updating the literature, including research published after 2015 (see references 4-11, 13, 23, 37, 40,41, 43, 44) and incorporating their suggested papers into our introduction (references 8,9,40) regarding adolescents’ views on parenting.
2. A short paragraph to outline and explain important classic parenting studies by Baumrind as well as Maccoby & Martin was added to lines 33-46.
3&9. We thank the reviewer’s suggestion regarding analysing the 5 Chinese parenting dimensions separately. However, in order to maintain the integrity of the parenting instruments used and to allow for comparisons across the three broad parenting styles (authoritative, authoritarian and Chinese) at an overall level, it would not be justifiable to breakdown the Chinese parenting style into 5 smaller subdimensions. We did appreciate the reviewer’s suggestions in providing more information on the Chinese parenting concepts and have elaborated on in further in lines 88-166.
4. The hypotheses were formulated and clarified in lines 272-276. The contributions of the study towards the existing literature are further expanded in lines 206-212.
5. More information regarding the sample, research ethics, and research procedures including the children’s age range, household income, and parental marital status have been given (see lines 279-311).
6. We added a sentence explaining why only one parent (the primary-care-giving parents) filled out the parenting questionnaires in lines 301-302. Although the numbers of the fathers and mothers were not balanced, the homogeneity tests in the MANCOVA did confirm the homogeneity between the fathers and mothers. We also ran separate regressions for fathers and mothers to compare the regression models and coefficients, and the results showed the same trends as the full sample, therefore eliminating the reviewers’ concerns about the distorted result by unbalanced sample group. Moreover, in the hierarchical regression models, parental gender was entered as one of the control variables in model 1, therefore the effect of parental gender is accounted for in the subsequent analyses (Model 2 and Model 3).
7. The results from confirmatory factor analysis regarding the perceived parenting scale was added to the measure description of perceived parenting scale (see lines 346-349 and Table 1), which confirmed the factor structure was maintained.
8. Example items for each measure were added to illustrate them in the method section (Lines 321-326; 334-339; and 362-365).
9. Addressed together with point 3.
10. Regarding the scores of The Brief Symptom Rating Scale (BSRS-5), we have added more description on its usage and scoring (see lines 368-371). As the nature of this scale (screening for psychological symptomologies) and the nature of the sample (geographically stratified representative normal sample), scores generally below 5 is expected. However, the pupils did score a full range (from 0-20, see Table 2), reflecting the nature of this measure. Moreover, the homoskedasticity assumption for the hierarchical analyses was not violated with our sample, which we checked with the residual plots. Therefore, we would like to reassure the reviewer that this has been taken into consideration.
Reviewer 3 Report
The present manuscript includes interesting and novelty results about the relationships between parent-reported parenting, child-perceived parenting and children’s mental health in Taiwanese children. This study shows some interesting differences between the perception of parents and children, and highlights the effect of the child-perceived parenting on children’s mental health. However, the review of previous studies about parenting is not sufficiently updated, and some aspects of this work should be improved.
Introduction
There are numerous recent studies on parenting and children’s wellbeing, including the analysis of some cultural differences in parental styles (for example, comparing European and South American countries). However, in this work only two studies are included with reference after 2014. The authors must update their references.
Also, there are many studies on adolescents' perception of parental styles of parenting, highlighting the relationship between this perception and their psychosocial well-being. It would be convenient to include these studies, since in a similar way to this work they emphasize the importance of the perception of the children of parenting styles.
Since the differential aspects of Chinese parenting are analyzed in this study, it would be interesting to extend the description of the characteristics of this parenting style in the introduction.
Methods
The research design is appropriate. However, it would be convenient to provide more information about the sample of parents: only a single parent could answer the questionnaires? there was the possibility of answering both parents? how many parents refused to participate in the study? Also, since the monthly family income variable is included in the analyzes, more information should be provided on the socioeconomic characteristics of the sample. In addition, should be described how the monthly family income was measured.
Regarding the Perceived Parenting Scale, the authors indicate that it is an adapted scale. It would be convenient to provide more information about how this adaptation was made and the results of the factor analysis of this scale. In relation to the reliability of this scale, the internal consistency for perceived Chinese parenting subscale (.574) is not high. This coefficient is low, and this could be a limitation.
Results:
(line 204): “The ANCOVA revealed significant effects for monthly family income (covariate) on parent-reported authoritative parenting (F (1, 621) = 13.398, p < .001), parent-reported authoritarian parenting (F (1, 621) = 4.530, p < .05) and child-perceived authoritative parenting (F (1, 621) = 15.077, p < .01)”. In relation to this result, it would be convenient provide more information about how different monthly family income are related to different parenting styles.
Discussion
The conclusions are supported by the results, and authors provide interesting data about the relationship between child-perceived parenting and children’s mental health in Taiwanese children.
As a suggestion, these results could also be related to the parent-child communication, since some differences in this communication have also been observed according to the gender of the children and the gender of the parent.
Moreover, it would be interesting to expand the discussion of results in relation to the characteristics of Chinese parenting.
Finally, the description of the practical implications of these results could be expanded.
Author Response
Reviewer 3
First of all, we would like to thank reviewer 3 for their constructive comments and valuable suggestions.
Please see our responses detailed in the following:
1. We have updated the literatures reviewed, including research published after 2015 (see references 4-11, 13, 23, 37, 40,41, 43, 44). And these updated studies included studies using adolescents as the respondents for parenting and mental health. Some more elaboration was given in lines 187 to196.
2. We thank the reviewer for this thoughtful suggestion. More information and elaboration on the Chinese parenting concepts have been added in lines 88-166.
3. We have provided more detailed reports on the sample characteristics and the information suggested by the reviewer. Please see lines 279-311 for more information regarding the sample, research ethics, and research procedures including the children’s age range, household income, and parental marital status.
4. The results from confirmatory factor analysis regarding the perceived parenting scale were added to the measure description of perceived parenting scale (see lines 346-349 and Table 1), which confirmed the factor structure was maintained. We have also added the relatively low internal consistency for perceived Chinese parenting subscale in the limitation (see lines 564-565).
5. The relationships between monthly family income and different parenting styles were already illustrated in the correlation analyses (see lines 391-394), therefore they were not further reiterated following the ANCOVA.
6. We thank the reviewer for the suggestion for discussing this in relevance to parent-child communication. Some discussions were added in lines 550-554.
7. Discussions in relation to Chinese parenting are expanded in lines 504-536.
8. Discussion of practical implications was further expanded in lines 579-584.Round 2
Reviewer 2 Report
I have reviewed the new version. I think the manuscript has improved but it has still serious problems.The authors have not yet advanced far enough satisfactorily to explain the point 3. What is the integrity of the parenting instruments? I know the Parenting Styles and Dimensions Questionnaire. This messure includes 3 parenting styles: Authoritative parenting style (subfactor 1. Connection Dimension -Warmth & Support-, subfactor 2. Regulation Dimension -Reasoning/Induction-, subfactor 3. Autonomy Granting Dimension -Democratic Participation-), Authoritarian parenting style (subfactor 1. Physical Coercion Dimension, subfactor 2. Verbal Hostility Dimension, subfactor 3. Non-Reasoning/Punitive Dimension), and Permissive parenting style (indulgent dimension). Where is the permisive parenting in this study? How have the authors included the Chinese-specific parenting dimension? Moreover, I believe the authors have difficulties in differentiatingbetween parenting dimension and parenting styles.
In the introduction, authors wrote about specific cultural and contextual variations in parenting but they only used chinesse parenting based in the shame dimension. The parenting chinesse style should not be considered only homogeneous parenting style. For instance, authors like Kim, Wang, Orozco-Lapray, Shen, & Murtuza (2013) explained eight dimensions and four parenting styles in Chinese American families.
Reviewer 3 Report
The authors have given an adequate response to the comments and have significantly improved the manuscript. The changes that were indicated have been adequately carried out. In my opinion, the revised manuscript can be published.
Only, minor changes are suggested to the authors:
- line 162: SD in italics
- line 164: SD in italics